# The Influence of Environmental Factors on Seed Germination of *Polygonum perfoliatum* L.: Implications for Management

Shahid Farooq [1,*], Huseyin Onen [2,*], Sonnur Tad [2], Cumali Ozaslan [3], Samy F. Mahmoud [4], Marian Brestic [5,6,*], Marek Zivcak [5], Milan Skalicky [6] and Ahmed M. El-Shehawi [4]

1   Department of Plant Protection, Harran University, Şanlıurfa 63200, Turkey
2   Department of Plant Protection, Gaziosmanpaşa University, Tokat 60240, Turkey; sonnurtad@gmail.com
3   Department of Plant Protection, Dicle University, Diyarbakir 21280, Turkey; cumaliz@yahoo.com
4   Department of Biotechnology, College of Science, Taif University, P.O. Box 11099, Taif 21944, Saudi Arabia; s.farouk@tu.edu.sa (S.F.M.); elshehawi@hotmail.com (A.M.E.-S.)
5   Department of Plant Physiology, Slovak University of Agriculture, 94976 Nitra, Slovakia; marek.zivcak@uniag.sk
6   Department of Botany and Plant Physiology, Faculty of Agrobiology, Food, and Natural Resources, Czech University of Life Sciences, Kamýcka 129, 165 00 Prague, Czech Republic; skalicky@af.czu.cz
*   Correspondence: csfa2006@gmail.com (S.F.); onenhuseyin@gmail.com (H.O.); marian.brestic@uniag.sk (M.B.)

**Abstract:** *Polygonum perfoliatum* L. is an aggressive vine, currently invading the Black Sea region, Turkey. However, information about the seed germination biology of this species is scanty. The objective of the current study was to determine the seed germination biology of three naturalized populations of this species. Chemical scarification with 98% sulfuric acid for 30 min followed by cold-wet stratification at 4 °C for 4 weeks effectively released seed dormancy in tested populations. Seeds of all populations required a 12 h photoperiod for the highest germination, while germination under continuous dark and light remained similar. The seeds were able to germinate under a wide range of constant (5–40 °C) and alternating temperatures, pH (3–11), osmotic potential (0 to −1.4 MPa) and salinity (0–500 mM NaCl). However, the peak germination was observed under 20 °C constant and 20/15 °C alternating day/night temperature, and pH 6.8. Seeds of all populations were able to withstand 200 mM salinity and −0.6 MPa osmotic potential. Increasing seed burial depth initially stimulated seedling emergence and then a sharp decline was observed for the seeds buried below >2 cm depth. More than 90% of the seeds were unable to emerge when buried >6 cm depth. *Polygonum perfoliatum* has a large potential for range expansion; therefore, immediate management of the naturalized populations is warranted. This weed species in agricultural fields can be managed by burying the seeds in deeper soil layers (6 cm), while post-emergence management strategies need to be developed for roadside populations.

**Keywords:** seed germination biology; *Polygonum perfoliatum*; seedling emergence; environmental factors; range expansion

## 1. Introduction

Biological invasions are impacting natural and agricultural ecosystems in the world, and invasive alien species are considered as one of the major threats to biodiversity [1]. The interest on alien plant species has increased during the last decades due to their increased economic and ecological impacts [2–4]. Therefore, identifying the traits promoting the invasion and naturalization of alien plant species in exotic ranges has been an important question in ecology [5,6]. Rapid dispersal followed by successful establishment determines the invasion success of alien plant species into new habitats [7,8]; however, dispersal and subsequent naturalization are significantly affected by climatic conditions of the invaded regions [9]. Invasion dynamics are significantly altered by the selection of superior traits, i.e., successful seed germination, seedling establishment, rapid growth and seed

dispersal under varying environmental conditions [10,11]. Seed germination, being the first step in the life history of plant species, strongly affects various aspects of plant biology, including population dynamics, geographical distribution and responses to climate change [12–15]. Thus, seed germination plays a vital role in the invasion success of invasive plant species [16] and studying seed germination traits could provide valuable insights into the species' invasiveness at regional and global scales.

Seed germination is vital for the persistence of many plant species [17] and readily affected by various environmental factors including light, temperature (either constant or alternating day/night), soil pH, soil salinity, water stress and seed burial depth [18–24]. The specific requirements (for all these environmental factors) of a species to complete essential physiological processes for seed germination provide valuable information regarding its possible invasion success in introduced regions [25,26]. Seed germination under adverse environmental conditions and broad seed germination niche are good indicators of invasion success [16,27–29]. Therefore, seed germination must be evaluated through experimental studies in laboratory or greenhouse, preferably in both [30].

Seed dormancy allows longer persistence in soil or to avoid unfavorable environmental conditions. Seed germination traits, including dormancy are driven by genetic adaptation and climate through maternal effects [31,32].

Seed germination responses are mutually dependent as light determines the ability to persist in soil seed bank as well as seedling emergence. Therefore, studies on seed germination biology/niche could determine the ecological response of the species to adverse environmental conditions such as soil salinity, low or high soil reactions, water stress, flooding, etc. [21,33–35]. Although literature on seed germination biology/niche exists for some exotic species in their native ranges, knowledge on their seed germination niche in introduced ranges is limited. Studying seed germination niche of naturalized populations in a new habitat different from their native ranges could provide valuable insights into range expansion potential, as well as help in estimating their future spread ranges.

Mile-a-minute weed or devil's tearthumb (*Polygonum perfoliatum* L., syn. = *Persicaria perfoliata* (L.) H. Gross) is an aggressive exotic vine native to Asia [36], which currently invades a large portion of the Black Sea region in Turkey [9,37–40]. Mile-a-minute completely shades native communities and creates monoculture of large patches in the region. The species invades agricultural areas and recently suspected as a lurking peril for the sustainability of tea plantations in the region [38]. Invasion is not only limited to tea plantations, rather extends to kiwi orchards, vegetable fields, hazelnut plantations, river sides, forest sides, roadsides and abandoned lands [37,39]. The species has also been considered a noxious invader in the United States and several studies have been conducted on its biological control [36,41]. Besides, extensive studies have been conducted on seed dormancy release [42–44], but not on seed germination biology. Although species invades a large portion of the Black Sea region, no information is available on seed germination niche of Turkish populations.

The current study was conducted to determine the seed germination niche of three naturalized Turkish populations of the species. We hypothesized that the species have a broad seed germination niche, which promoted its invasion success in the Black Sea region, Turkey and could facilitate further range expansion in the country. To test this hypothesis, we conducted seed germination tests under different light/dark regimes, constant and alternating day/night temperatures, different levels of pH, salinity and osmotic stress, and observed seedling emergence from various seed burial depths. The results would help in estimating the future spread potential of the species in the country and provide insights for possible management options.

## 2. Materials and Methods

### 2.1. Seed Collection

The achenes (hereafter seeds) of Mile-a-minute were collected from three different populations namely Borçka, Ardeşen and Dernekpazarı (Table 1), which are the names of

small towns situated in Artvin, Rize and Trabzon provinces of the eastern Black Sea region, Turkey, respectively. The seeds were randomly collected from fifty mother plants during 2016 and brought to laboratory. Seeds were dried for two weeks under shade to meet after-ripening requirements for germination [44,45]. The dried seeds were then cleaned and used in the experiments. The geographic locations and climatic conditions prevailing at the sites of origin of tested populations are given in Table 2.

**Table 1.** Physico-chemical properties of the soils of different *Polygonum perfoliatum* populations included in the current study.

| Location | Clay | Sand | Silt | CaCO$_3$ | OM | pH | EC | P |
|---|---|---|---|---|---|---|---|---|
| | % | | | | | | μS cm$^{-1}$ | mg kg$^{-1}$ |
| Borçka | 34.50 | 53.20 | 12.30 | 1.16 | 2.21 | 6.98 | 92.90 | 1.82 |
| Ardeşen | 32.00 | 54.00 | 14.00 | 3.54 | 3.94 | 7.02 | 458.50 | 7.26 |
| Dernekpazarı | 29.00 | 57.00 | 14.00 | 1.30 | 1.74 | 7.04 | 165.00 | 8.74 |

OM = organic matter, EC = electrical conductivity, P = plant available phosphorus.

**Table 2.** Geographic location and climatic data of different *Polygonum perfoliatum* populations included in the current study.

| Location | Lat | Long | Province | Altitude (m) | RH | T$_{ave}$ | T$_{max}$ | T$_{min}$ | Rainfall | PET |
|---|---|---|---|---|---|---|---|---|---|---|
| | | | | | (%) | | (°C) | | (mm) | |
| Borçka | 41.40 | 41.50 | Artvin | 173 | 61.82 | 12.18 | 18.35 | 11.52 | 1244.01 | 744.66 |
| Ardeşen | 41.15 | 41.01 | Rize | 6 | 62.29 | 13.25 | 17.62 | 10.53 | 1620.38 | 720.34 |
| Dernekpazarı | 40.91 | 40.28 | Trabzon | 190 | 68.55 | 13.45 | 18.17 | 10.15 | 1947.65 | 750.63 |

RH = relative humidity, T$_{ave}$ = average annual temperature, T$_{max}$ = maximum annual temperature, T$_{min}$ = minimum annual temperature, PET = potential evapotranspiration.

### 2.2. Evaluation of Seed Dormancy Status and Seed Dormancy Release

The dormancy level of the seeds was evaluated first, which indicated that seeds of all populations were extremely dormant. Seed dormancy of Mile-a-minute is both coat imposed (hard water impermeable pericarp) and due to inhibiting substances; therefore, chemical scarification with 93% sulfuric acid followed by 4 weeks cold-wet stratification at 4 °C is needed for effective release of seed dormancy [44]. We used concentrated sulfuric acid (98%) and intended to know whether a higher concentration of sulfuric acid reduces the scarification or stratification duration required to release the dormancy. Thus, we included two different chemical scarification durations, i.e., 30 and 60 min followed by two cold-wet stratification (at 4 °C) durations, i.e., 2 and 4 weeks. Chemical scarification and cold-wet stratification alone or in combination were used in the study, which made total of six different seed dormancy release treatments. A control treatment without scarification and stratification was also included in the experiment. Seeds (500) were dipped either for 30 or 60 min in 98% sulfuric acid, thoroughly rinsed with distilled water, surface dried with filter paper and subjected to cold-wet stratification at 4 °C for 2 and 4 weeks. Seeds (50 in each Petri dish) were placed in 9-cm Petri dishes between moistened filter paper layers, sealed with paraffin film and incubated under complete dark conditions for the required stratification duration according to the treatments. The germination of chemically scarified and cold-wet stratified seeds was then evaluated according the procedure described below.

### 2.3. General Procedure for Germination Tests

Seed germination of non-dormant seeds was observed by evenly placing 50 seeds in 9-cm Petri dishes having 2 layers of Whatman No. 1 filter paper. The filter paper layers were moistened either with distilled water or treatment solution (for pH, salinity and osmotic potential experiments) following the procedures described by Chauhan [20] and Fang et al. [46]. The Petri dishes were incubated in 12 h photoperiod (except light and dark experiments) and 20 °C temperature (except constant temperature experiment)

found optimum for seed germination of all populations in temperature experiment. Light was supplied in the incubators with fluorescent lamps having photosynthetic photon flux density of 140 lmol m$^{-2}$ s$^{-1}$. Seed germination was observed daily (except dark treatment where germination was noted after 30 days of incubation period) and germinated seeds were removed from the Petri dishes. The places of Petri dishes within the incubators were changed daily to avoid any bias and moisture was supplied with distilled water as needed. Germination counts were terminated after 30 days of incubation period, final germination percentage was counted and used for the interpretation of results. All experiments were conducted according to factorial design keeping populations as main factor, while treatments were taken as sub-factor. Every treatment in each experiment had five replications (two Petri dishes were considered as one replication) and all experiments were repeated (i.e., two experimental runs for each experiment).

### 2.4. Impact of Constant Temperatures on Seed Germination

The non-dormant seeds of all populations were incubated in nine different constant temperatures (5, 10, 15, 20, 25, 30, 35, 40 and 45 °C) to infer the effects of temperature on seed germination. The temperatures were selected to cover the full temperature range prevailing in the country.

### 2.5. Impact of Alternating Day/Night Temperatures on Seed Germination

The non-dormant seeds of all populations were incubated in seven different alternating day/night temperatures (10/10, 15/10, 15/15, 20/15, 25/20, 30/25 and 30/30 °C) to infer the effects of alternating day/night temperature on seed germination. The alternating temperatures were selected based on the constant temperature resulting in the highest seed germination.

### 2.6. Effect of Light/Dark on Seed Germination

Non-dormant seeds of all populations were incubated in three different light/dark periods, i.e., 12 h photoperiod, continuous light and continuous dark to observe the effects of light on seed germination. The Petri dishes of complete dark treatment were wrapped in 3 layers of aluminum foil to exclude the effects of light on seed germination.

### 2.7. Impact of pH on Seed Germination

The non-dormant seeds of all populations were incubated under different pH levels to test the effect of pH on seed germination. Germination was observed under nine different pH levels, i.e., 3, 4, 5 and 6 (for acidic medium), 7 (distilled water for neutral medium), and 8, 9, 10 and 11 (for alkaline medium). Buffer solutions were prepared following Chachalis and Reddy [47] for attaining the desired pH levels.

### 2.8. Impact of Osmotic and Salt Stress on Germination

Non-dormant seeds of all tested populations were incubated under different salinity and osmotic potential levels to test their effects on seed germination. Seeds were incubated under eight different osmotic potentials (−0.2, −0.4, −0.6, −0.8, −1.0, −1.2, −1.4 and −1.6 MPa) along with a control treatment (0 MPa, distilled water). The aqueous solutions with desired osmotic potentials were prepared by dissolving polyethylene glycol 6000 in distilled water following Michel and Kaufmann [48]. Similarly, seeds were incubated under eight different salinity levels (50, 100, 150, 200, 300, 400, 500 and 600 mM sodium chloride) along with control treatment (0 mM, distilled water). The calculated amount of sodium chloride (NaCl) was dissolved in distilled water to achieve the treatment solutions of desired salinity level.

### 2.9. Effect of Burial Depth on Seedling Emergence

Seedling emergence was observed in plastic pots (25 cm diameter) in a greenhouse experiment. Twenty non-dormant seeds of all populations were either placed at surface

or buried at various depths (0.5, 1, 1.5, 2, 4, 6, 8, 10 and 12 cm) to observe the effects of seed burial depth on seedling emergence. The greenhouse was maintained at $25 \pm 2\,°C$ day and $20 \pm 2\,°C$ night temperature and 12 h photoperiod (300 to 500 lmol m$^{-2}$ s$^{-1}$). The pots were initially sub-irrigated and then surface irrigated daily to maintain the pots at field capacity [49]. The soil used in the experiment was taken from the research farm of the Gaziosmanpaşa University, Tokat province, Turkey having no previous seed bank of Mile-a-minute. Soil was autoclaved before using in the experiment. Seedling emergence were recorded for 40 days taking cotyledon appearance as criterion.

### 2.10. Statistical Analysis

The final germination and seedling emergence data were used in statistical analyses. The difference between experimental runs were tested first using paired t test, which were non-significant. Thus, the data of both experimental runs were pooled for further analyses. Two-way Fisher's analysis of variance (ANOVA) was used to test the significance among populations and treatments in all experiments [50]. In case ANOVA denoted significant differences, means of light/dark, alternating day/night temperature and dormancy experiments were separated using least significant difference post-hoc test at 5% probability level.

Two different types of models were fitted on the final germination percentage data of constant temperature, pH, seed burial depth and osmotic and salinity stress experiments.

The three-parameter sigmoid model was fitted to the final germination percentage data of osmotic and salinity stress experiments. The model was

$$G = G_{max}/(1 + e\,[-x - T_{50})/G_{rate}], \tag{1}$$

where G is the cumulative percentage germination at time x, $G_{max}$ the maximum germination (%), $T_{50}$ osmotic potential or salinity level required to inhibit 50% of maximum germination, and $G_{rate}$ the slope.

Similarly, the data of constant temperatures, pH and seed burial experiments were subjected to a three-parameter Gaussian model. The model was:

$$G = a \times e\,[-0.5 - \{(x - b)/c\}2] \tag{2}$$

The Gaussian model gives a "bell curved" graph. In the above model, "a" corresponds to the height of the curve's peak (maximum germination or seedling emergence), "b" the position of center of the peak (temperature, pH or depth of seed burial to achieve maximum germination or seedling emergence) and "c" width of the "bell".

The ANOVA was performed by using SPSS version 21.0 (IBM Corp: Armonk, NY, USA) [51], while sigmoid and Gaussian models were fitted on SigmaPlot version 13.0 (Systat Software, San Jose, CA, USA).

## 3. Results

### 3.1. Seed Dormancy Release

Seed dormancy was significantly ($p \leq 0.01$) influenced by seed dormancy release treatments, while populations and interactions among populations and seed dormancy release treatments were non-significant ($p > 0.05$) (Table S1). The lowest seed germination (no germination) was observed in control treatment (no scarification and stratification), while scarification for 30 min followed by 4 weeks cold-wet stratification resulted in the highest final germination (i.e., seed dormancy release) (Figure 1).

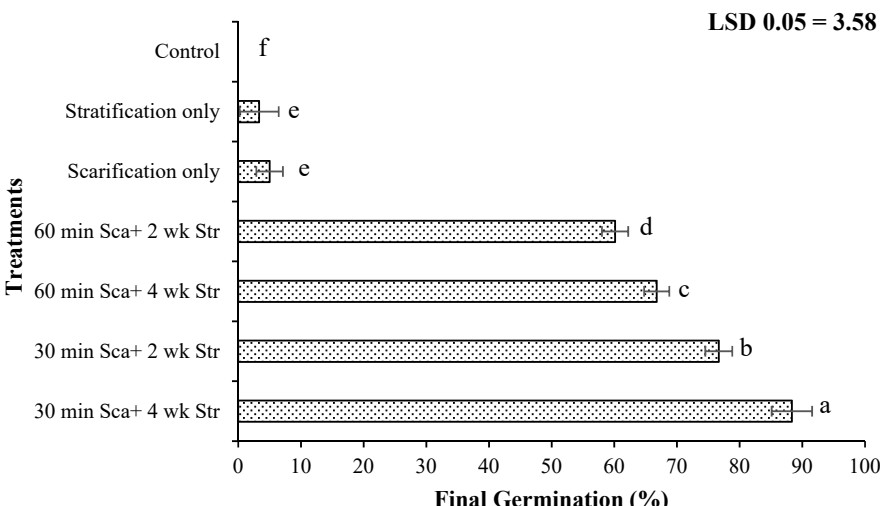

**Figure 1.** The effect of different seed dormancy release treatments on seed dormancy release (in terms of final germination) of *Polygonum perfoliatum* (The abbreviated words in *Y*-axis are; Sca = chemical scarification, Str = stratification, min = minutes, wk = weeks). Means sharing the same letters are statistically non-significant (*p* > 0.05), while horizontal bars represent standard errors of the means.

### 3.2. Light/Dark Regimes

The tested light/dark period significantly ($p \leq 0.01$) affected the germination, while the effects of populations and interaction of populations and light/dark period were non-significant (Table S2). The highest germination (80.83%) was recorded under 12 h photoperiod, whereas continuous light and continuous dark resulted in 20% lower germination (Figure 2).

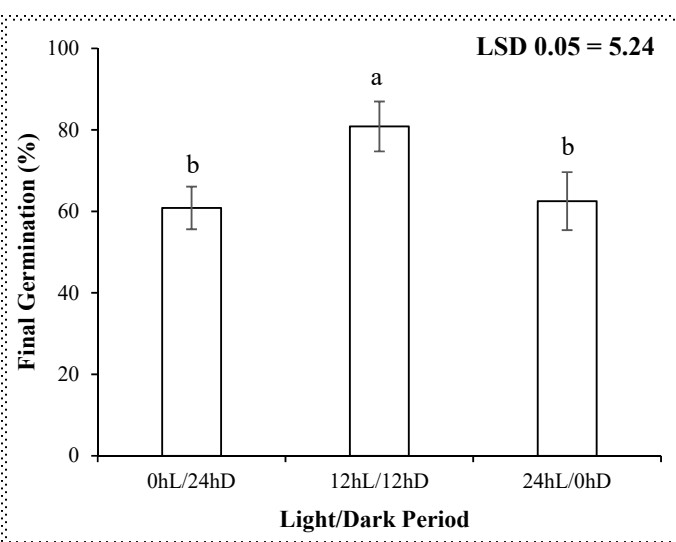

**Figure 2.** The effect of different light/dark regimes on final germination percentage of *Polygonum perfoliatum.* Means sharing the same letters are statistically non-significant (*p* > 0.05), while vertical bars represent standard errors of the means.

### 3.3. Constant Temperatures

Seed germination was significantly influenced by populations ($p \leq 0.05$), constant temperatures ($p \leq 0.01$) and their interactions ($p \leq 0.05$) (Table S3). Seed germination was stimulated by increasing temperature, reached to the peak and then a constant decline in germination was noted (Figure 3). There were almost no differences among tested populations regarding optimum temperature for peak germination (19.29, 20.67 and 20.87 °C for Borçka, Ardeşen and Dernekpazarı, respectively).

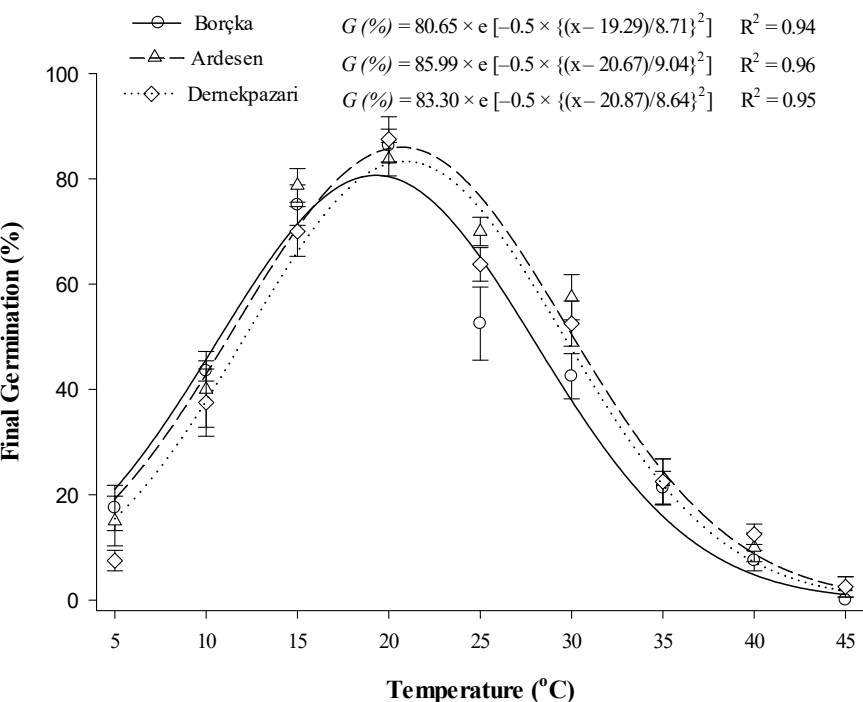

**Figure 3.** Effect of different constant temperatures on the final germination (%) of three *Polygonum perfoliatum* populations incubated in 12 h light and dark period. The vertical bars are standard errors of means, while the lines represent 3 parametric Gaussian model fitted to the final germination percentage data obtained after 30 days incubation period.

### 3.4. Alternating Temperatures

Different alternating temperatures, populations and their interaction significantly ($p \leq 0.05$) altered seed germination (Table S4). The lowest seed germination of all populations was recorded for 10/10 °C alternating temperatures, while all populations observed the highest seed germination under 20/15 °C (Figure 4).

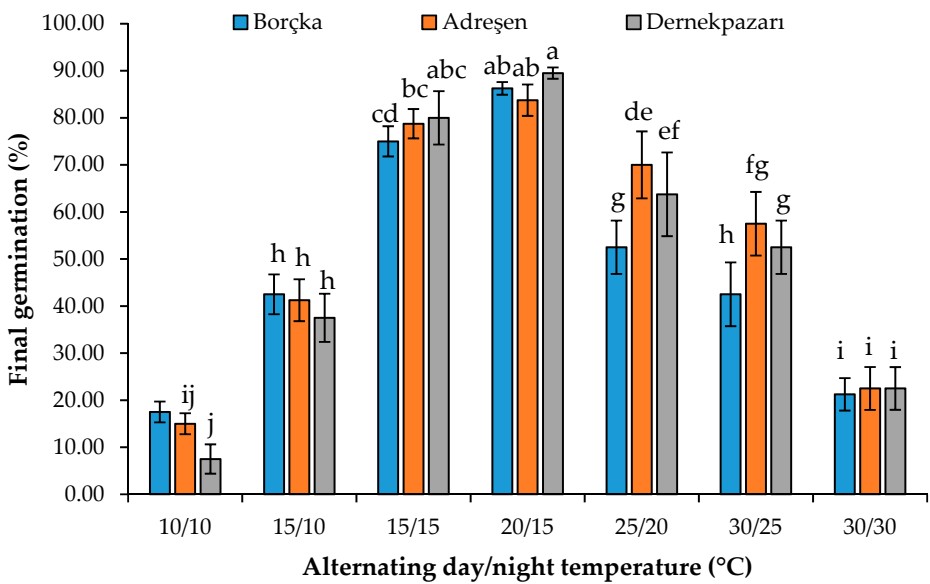

**Figure 4.** Effect of different alternating day/night temperatures on the final germination (%) of three *Polygonum perfoliatum* populations incubated in 12 h light and dark period. The vertical bars are standard errors of means, while the means having different letters are statistically significant ($p \leq 0.05$).

### 3.5. Soil Reaction/pH

The pH levels and populations × pH interactions had significant ($p \leq 0.01$) effect on seed germination (Table S5). Seeds of all populations germinated under all pH levels included in the study; however, the highest germination percentage was observed under near to neutral pH (Figure 5). The pH levels required to reach the peak germination were 6.27, 6.75 and 6.51 for Borçka, Ardeşen and Dernekpazarı populations, respectively.

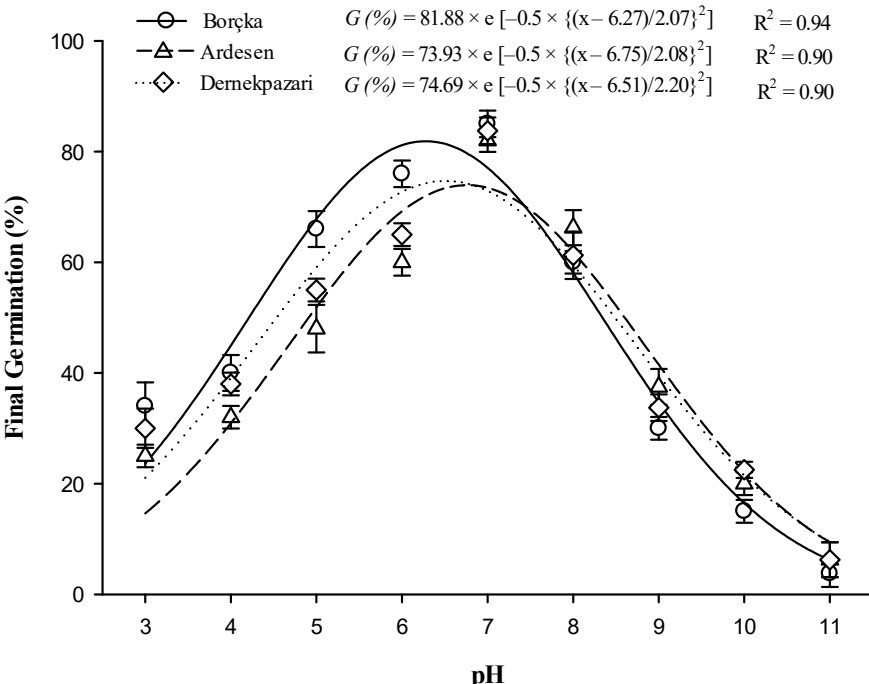

**Figure 5.** Effect of different pH levels on the final germination (%) of three *Polygonum perfoliatum* populations incubated in 12 h light and dark period and 20 °C constant temperature. The vertical bars are standard errors of means, while the lines represent 3 parametric Gaussian model fitted to the final germination percentage data obtained after 30 days incubation period.

### 3.6. Salinity and Osmotic Potential

Seed germination was significantly affected by populations, salinity levels and osmotic potential levels, while interactions of populations with salinity and osmotic potential levels had non-significant effect (Tables S6 and S7). A sharp decline was observed in seed germination percentage with increasing salinity and osmotic potential up to 400 mM salinity level and −1.2 MPa osmotic potential (Figures 6 and 7). Salinity levels which decreased 50% of the maximum germination were 196.09, 229.03 and 183.43 mM for Borçka, Ardeşen and Dernekpazarı populations, respectively (Figure 6). Similarly, the osmotic potentials required to inhibit 50% of the maximum germination were −0.64, −0.69 and −0.57 MPa for Borçka, Ardeşen and Dernekpazarı populations, respectively (Figure 7). The seeds of Ardeşen population proved relatively tolerant to higher salinity level compared with Borçka and Dernekpazarı populations, which could be attributed to soil properties at the sites of origin (Table 1).

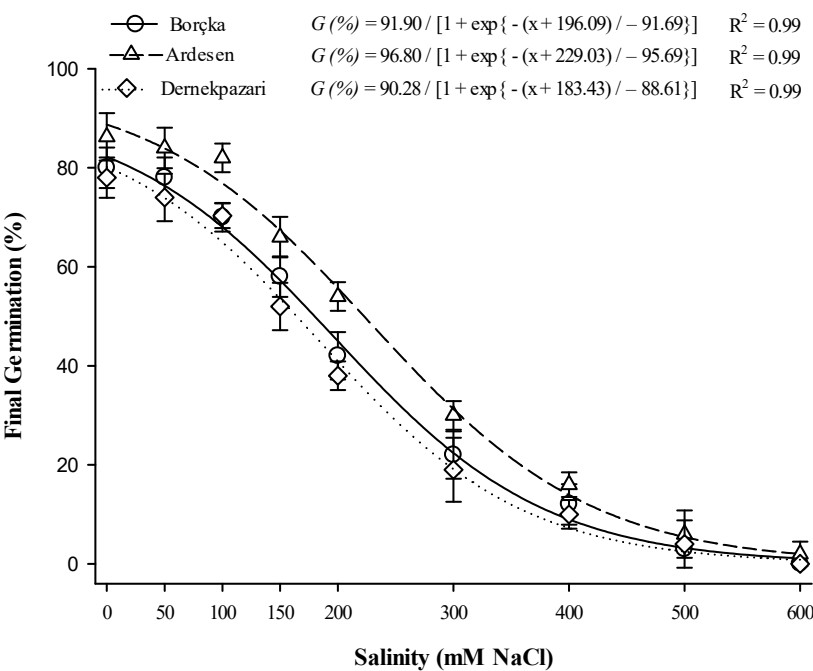

**Figure 6.** Effect of different salinity levels (mM) on the final germination (%) of three *Polygonum perfoliatum* populations incubated in 12 h light and dark period and 20 °C. The vertical bars are standard errors of means, while the lines represent 3 parametric sigmoidal model fitted to the final germination % data obtained after 30 days of initiation of the experiment.

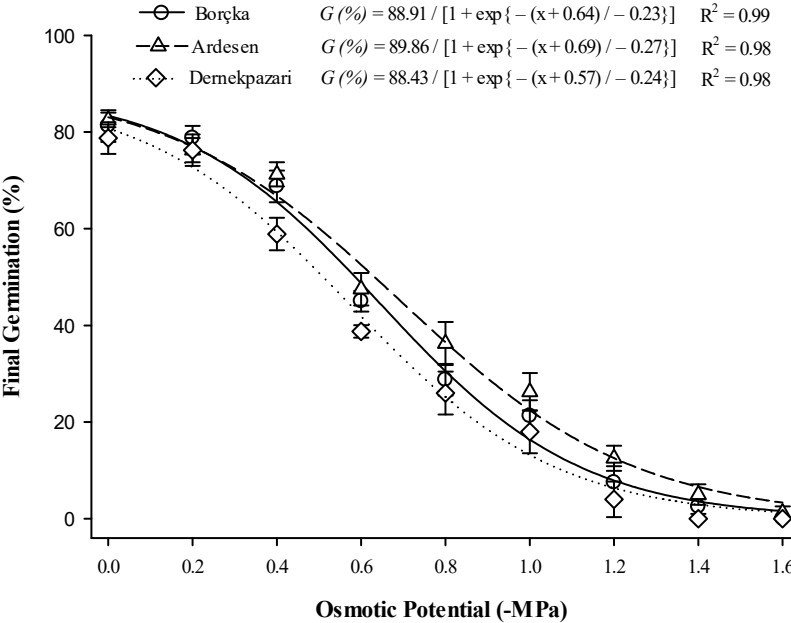

**Figure 7.** Effect of different osmotic potentials (-MPa) on the final germination (%) of three *Polygonum perfoliatum* populations incubated in 12 h light and dark period and 20 °C. The vertical bars are standard errors of means, while the lines represent 3 parametric sigmoidal model fitted to the final germination % data obtained after 30 days of initiation of the experiment.

### 3.7. Seedling Emergence

Different seed burial depths significantly ($p \le 0.01$) influenced the seedling emergence (Table S8). Seedling emergence was initially increased up to 2 cm burial depth and then a sharp decline was observed (Figure 8). Seedling emergence was 10% for the seeds placed on surface, reached to 70% for the seeds buried 2 cm deep, reduced to 40% at 4 cm and

lowered to 4.16% under 10 cm burial depth. None of the seeds were able to emerge when buried >10 cm deep. The optimum seed burial depth for maximum seedling emergence was 2.56, 2.53 and 2.53 cm for Borçka, Ardeşen and Dernekpazarı populations, respectively.

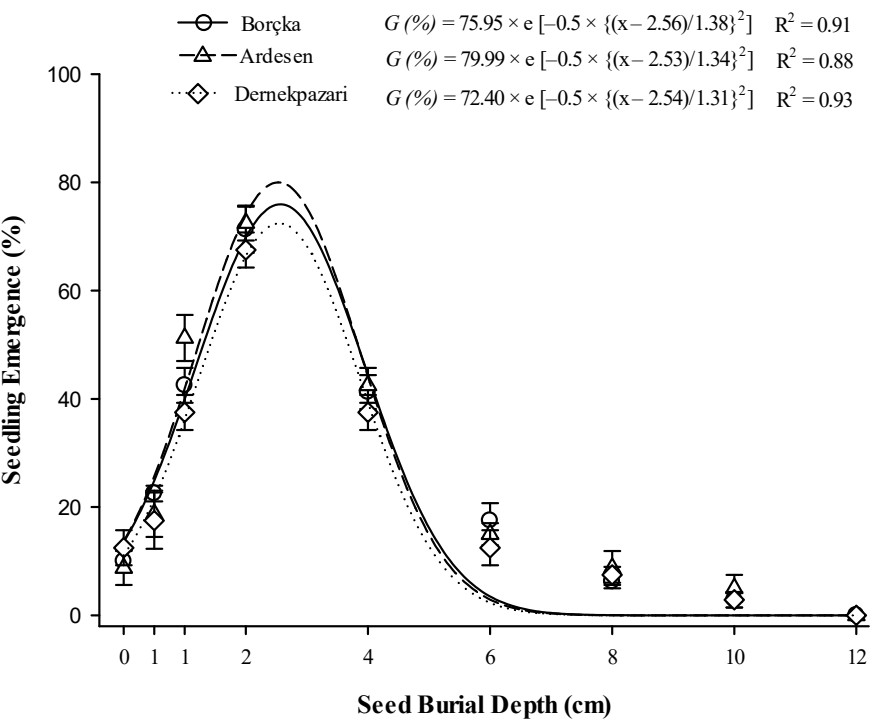

**Figure 8.** Effect of different seed burial depths on seedling emergence (%) of three *Polygonum perfoliatum* populations. The vertical bars are standard errors of means, while the lines represent 3 parametric Gaussian model fitted to the final germination percentage data obtained after 30 days incubation period.

## 4. Discussion

The seeds of Mile-a-minute were capable of germinating under various constant and alternating day/night temperatures, pH levels, salinity and osmotic stress levels and able to emerge from various seed burial depths, which suggested that it has a broad seed germination niche. The breadth of seed germination niche helped the species to germinate and establish in the current distribution range. These results indicate that species would further become increasingly problematic in humid regions of the country like Black Sea region as indicated by Farooq et al. [9] and Önen et al. [37].

Seed dormancy is induced by different biotic or abiotic factors either internal or external to the seed coat [52]; thus, dormancy status of the seeds continuously varies and reaches to maximum or minimum at various timings of seed's life [53]. Therefore, knowledge of seed dormancy is crucial to understand the biology and ecology of plant species, as well as to predict the timing of seedling emergence [54]. Seed dormancy status/level is primarily determined by the environmental conditions faced by the maternal plants at the time of seed production [55]. Therefore, seeds of different individuals, even within same population, differ in their level of dormancy [53]. Cold-wet stratification followed by chemical scarification effectively releases seed dormancy of Mile-a-minute [42,56]. The results of current study are slightly different from those of Colpetzer and Hough-Goldstein [44] who suggested that chemical scarification for 60 min followed by 4 weeks cold-wet stratification effectively releases seed dormancy of Mile-a-minute. These differences could be explained with the concentration of sulfuric acid used in the current study for chemical scarification. Colpetzer and Hough-Goldstein [44] used 93% sulfuric acid, while in the current study 98% sulfuric acid was used. Thus, the concentration of the acid probably reduced the time required for chemical scarification. The findings of the current study suggest that

increasing the concentration of sulfuric acid can lower the chemical scarification time required for releasing the seed dormancy of Mile-a-minute. The cold-wet stratification probably increased the concentration of unsaturated fatty acids in cell membranes, which improved the membrane permeability, thus helped in releasing the seed dormancy [57].

The results indicate that the seeds of Mile-a-minute could germinate under different light and dark regimes; however, 12 h photoperiod significantly stimulated the germination compared to 0 and 24 h photoperiod. It has been contrarily suggested that light had no effect on the seed germination of Mile-a-minute populations in Korea [56]. Our results for 24 h photoperiod are in agreement with Yang and Kim [56], while slight differences could be explained with genetic variation among populations arising from different regions and subsequent ecological adaptations.

Temperature is an inherent requirement for seed germination as number of enzyme activities affecting hormone synthesis are influenced by temperature [18,19]. Seeds of all populations were capable of germinating under wide range (5–40 °C) of constant and alternating day/night temperatures. The results are in accordance with Johnson [42] who reported that seeds of Mile-a-minute could germinate under a wide range of temperatures (5–20 °C). However, our results indicated that the seeds are able to germinate under even wider range, i.e., 5–40 °C of temperature indicating possible adaptations to new environments. Germination ability over wide range of temperature has been suggested as an important determinant of the species persistence under novel environmental conditions [16,54].

Seed germination was linearly increased with increasing pH levels up to neutral pH and then a linear decline was observed. Our results are in good agreement with Okay [43] who concluded that seeds of Mile-a-minute can germinate under pH level of 3.5 without stratification; however, the highest germination occurs under neutral pH (i.e., 7.5). Higher germination under lower pH compared with higher pH can be attributed to the maternal effects [16,32] in current distribution range of the species where pH level is lower compared to the other parts of the country [58].

Seed germination traits are important determinants of invasion success as species could explore novel environments to extend its invasion range in Turkey. However, highly sensitive nature of seedlings to water and salinity stress [9] will probably limit range expansion to arid and saline areas. Although arid and saline areas are suspected to be safe, irrigated-semi-arid areas of the country with low salinity levels are under the risk of invasion. Therefore, seed transport to these areas by different vectors such as birds and humans [44] must be checked through different strategies, including quarantine and early warnings. Moreover, effective management strategies need to be developed for controlling/eradicating the species from current invasion range in the country. Farooq et al. [9] recently suggested that seedlings of Mile-a-minute are highly sensitive to salinity and water stress. Our results also indicate that the seed germination of Mile-a-minute is sensitive to increased salinity and osmotic potential. This sensitivity could be explained with the presence of the species in most humid region of the country having lower salt levels (Table 1: Sensoy et al. [59]).

The seeds placed on surface were unable to absorb sufficient moisture for germination; thus, lower emergence from soil surface can be explained with poor soil and seed contact resulting in low water imbibition. The Black Sea region receives the highest rainfall in the country, soil is mostly wet which may disturb $CO_2$-oxygen exchange in deeper soil layers [60], consequently lowering the seedling emergence from deep buried seeds. Moreover, lack of enough energy could be the other reason for lower emergence from deeper burial depths. The seeds of Mile-a-minute can remain viable for >6 years in soil seed bank [41–44]; thus, the lower emergence from surface and deep buried seeds could be a strategy opted by the species for longer persistence in the soil seed bank.

## 5. Conclusions

Our hypothesis pertaining to broad seed germination niche is valid for the areas having sufficient moisture, lower salinity and pH levels, which could be validated with the climatic and soil conditions of the entire Black Sea region and partially for Marmara and Aegean regions of Turkey. Keeping in view the results of current study modelling studies focusing the potential distribution under future climatic conditions are needed in the country.

**Supplementary Materials:** The following are available online at https://www.mdpi.com/article/10.3390/agronomy11061123/s1, Table S1: Two-way analysis of variance ofeffects of *Polygonum perfoliatum* populations, seed dormancy release treatments and their interactions on final germination percentage, Table S2: Two-way analysis of variance of effects of *Polygonum perfoliatum* populations, light dark regimes and their interactions on final germination percentage, Table S3: Two-way analysis of variance of effects of *Polygonum perfoliatum* populations, constant temperatures and their interactions on final germination percentage, Table S4: Two-way analysis of variance of effects of *Polygonum perfoliatum* populations, alternating temperaturesand their interactions on final germination percentage, Table S5: Two-way analysis of variance of effects of *Polygonum perfoliatum* populations, pH levels and their interactions on final germination percentage, Table S6: Two-way analysis of variance of effects of *Polygonum perfoliatum* populations, salinity levels and their interactions on final germination percentage, Table S7: Two-way analysis of variance of effects of *Polygonum perfoliatum* populations, different osmotic potentials and their interactions on final germination percentage, Table S8: Two-way analysis of variance of effects of *Polygonum perfoliatum* populations, seed burial depths and their interactions on seedling emergence.

**Author Contributions:** Conceptualization, H.O. and C.O.; data curation, S.F.; formal analysis, S.T.; funding acquisition, H.O., C.O., M.B. and A.M.E.-S.; investigation, S.F.M.; methodology, S.F.; resources, H.O., C.O. and A.M.E.-S.; software, S.F.; supervision, H.O.; visualization, S.F., S.T. and S.F.M.; writing—original draft, S.F.; writing—review and editing, H.O., S.T., C.O., A.M.E.-S., M.B., M.Z., M.S., and S.F.M. All authors have read and agreed to the published version of the manuscript.

**Funding:** This study was supported by the Scientific and Technological Council of Turkey (TUBITAK) with Grant Number 113 O 790 as a part of COST Action (TD 1209-European Information System for Alien Species). The current work was funded by Taif University Researchers Supporting Project number (TURSP-2020/138), Taif University, Taif, Saudi Arabia. This work was supported by the project EPPN2020-OPVaI-VA-ITMS313011T813.

**Institutional Review Board Statement:** Not applicable.

**Informed Consent Statement:** Not applicable.

**Data Availability Statement:** All data are within the manuscript and Supplementary Files.

**Conflicts of Interest:** The authors declare no conflict of interest. The funders had no role in the design of the study; in the collection, analyses, or interpretation of data; in the writing of the manuscript, or in the decision to publish the results.

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
