# Peer review of "The Influence of Environmental Factors on Seed Germination of Polygonum perfoliatum L.: Implications for Management"

_agronomy, doi:10.3390/agronomy11061123_

Round 1

Reviewer 1 Report

The aim of the research has been clearly stated. Appropriate research methods have been applied. Material and methods have been described clearly. The results obtained have been described sufficiently. However, Table 2 has been not included in the manuscript. The discussion is comprehensive and the correct conclusions have been drawn. The adequate references are included. Some detailed remarks I included in the manuscript.

Author Response

Reviewer 1:

Comment: The aim of the research has been clearly stated. Appropriate research methods have been applied. Material and methods have been described clearly. The results obtained have been described sufficiently. However, Table 2 has been not included in the manuscript. The discussion is comprehensive and the correct conclusions have been drawn. The adequate references are included. Some detailed remarks I included in the manuscript.

Response: Thank you for reviewing the manuscript and providing a constructive and positive feedback. We have corrected all the highlighted places in the manuscript. The wrongly cited references have been cited correctly. Table 2 has been included in the revised manuscript. We hope that you will recommend acceptance of the manuscript if the revisions are satisfactory.

Reviewer 2 Report

In Materials and methods:

Seed collection: Can author add GPS id for sampling area and collection time

Impact of constant temperature: Would you explain a little bit what you did choose these temperature ranges.  It is also important to see the impact of the alternate (day/night) temperature effect. This objective was missed here in this study. Can you explain why?

Author Response

Reviewer 2:

Comment: Seed collection: Can author add GPS id for sampling area and collection time

Response: The GPS coordinates of seed collection sites and collection time with other relevant information are given in Table 2.

Comment: Impact of constant temperature: Would you explain a little bit what you did choose these temperature ranges. 

Response: We intened to cover the full rage of Turkey weather prevailing at different regions to assess whether the species can germinate over there or not.

Comment: It is also important to see the impact of the alternate (day/night) temperature effect. This objective was missed here in this study. Can you explain why?

Response: We assessed the impact of alternating temperature as well. The data of alternating temperature is included in the revised manuscript.

Reviewer 3 Report

The topic is of wrathful consideration for weed scientists and agronomists. This is based upon effective weeds of that region. The write-up of whole manuscript is up to mark. English quality is acceptable but some minor spell check is needed. Upon the topic of the manuscript, M & M and Results are given in a proper and acceptable way. However, I believe that this study is somehow incomplete so that open field investigations could be effective for obtaining more valid findings that to be attractive and efficient for agronomists and weed scientists. Low novelty & effectiveness found in the study. 

Author Response

Reviewer 3:

Comment: The topic is of wrathful consideration for weed scientists and agronomists. This is based upon effective weeds of that region. The write-up of whole manuscript is up to mark. English quality is acceptable but some minor spell check is needed.

Response: Thank you for the positive feedback.

Comment: Upon the topic of the manuscript, M & M and Results are given in a proper and acceptable way. However, I believe that this study is somehow incomplete so that open field investigations could be effective for obtaining more valid findings that to be attractive and efficient for agronomists and weed scientists. Low novelty & effectiveness found in the study. 

Response: We are agree with the reviewer that field studies strengthen the findings of lab studies. However, this was the newly introduced weed in Turkey and nothing was known about the seed germination ecology. We were interested to determine the seed germination biology so that the field experiment can be conducted in the future. Please consider this as a limitation of the study.

Round 2

Reviewer 3 Report

Accept